# "Damn the Empire!": Imperial Excess, National Nostalgia, and Metaphysical Modernism in the Poetics of *Parade's End*

Molly Elizabeth Porter

Department of English, College of Arts and Sciences, University of Washington, Seattle, WD 98195-4330, USA; mep430@uw.edu

**Abstract:** Ford Madox Ford famously intended his First World War tetralogy *Parade's End* to have "for its purpose the obviating of all future wars". But why do we engage in war to begin with? Modernist literature provides some provocative explanations. Ford's Sylvia Tietjens, for example, proclaims that "You went to war when you desired to rape innumerable women. It was what war was for". And in the very same year, Virginia Woolf's shell-shocked Septimus Smith "went to France to save an England which consisted almost entirely of Shakespeare..." I argue that Ford's understanding of the causality of war involves a strange combination of these two explanations in *Parade's End*'s triangulation of seventeenth-century English literary tradition along with sexual and imperial conquest. While countless modernist novels exhibit a sensibility to the power of early modern poetry amidst battle, *Parade's End* displays a particularly emphatic and extended focus on the relationship between poetic tradition and war. Soldiers of various ranks "talk...in intimate undertones about the resemblances between the Petrarchan and the Shakespearean sonnet form", host timed sonnet competitions in the trenches, recurrently quote the seduction poetry of Marvell, and fantasize about George Herbert's lifespan being "the only satisfactory age in England...yet what chance had it today? Or, still more, to-morrow?". To answer this question, my own transtemporal study will use early modern scholarship to investigate seventeenth-century metaphysical poetry's dual power to inspire and potentially obviate war. Much has been written on this tetralogy's anti-linear plot but less on the broader temporality of its politico-literary vision. I contend that the metaphysical allusions of this text help Ford to show us the complexities of nationalism in the imperial conquest and imperial damnation that (early) modern aesthetics can catalyse.

**Keywords:** empire; modernism; early modern; poetics; First World War; Ford Madox Ford; George Herbert; Andrew Marvell

## 1. Prologue: A Meeting with Time

> O, how I long to travel back,
> And tread again that ancient track![1]
> But at my back I always hear
> Time's winged chariot hurrying near;[2]
> Meeting with Time, slack thing, said I,[3]
> What did the Lamb that He should die?[4]
>
> —Some "metaphysical" poets, muddled

My paper begins in an epigraphic experiment of confusion. I have pieced together this epigraphic collage of old seventeenth-century poets worried about growing old, for I like to imagine that my subject, the disorienting modernist innovator and early modern poetry appreciator Ford Madox Ford, would potentially approve. Midway through Ford's World War One tetralogy *Parade's End* (published 1924–1928[5]), for example, amid antilinear plots, elliptically[6] obfuscated revelations, and allusive jumps in time, the volunteer soldier

protagonist's commander General Campion seemingly speaks for the wearied, confused reader when he complains, "What is language for? What the *hell* is language for? We go round and round" (Ford 2012, p. 529)[7]. The General's own language certainly does, circling from modern trench warfare tactics to three-hundred-year-old seduction poetry. Earlier in this same conversation, General Campion extensively quotes a classic carpe diem poem by Andrew Marvell from memory in order to illustrate the precarity of "a general's life in this accursed war" (Ford 2012, p. 512). He goes on to explain: "You think all generals are illiterate fools. But I have spent a great deal of time in reading, though I never read anything written later than the seventeenth century" (Ford 2012, p. 512). While the narrator describes this moment of poetic allusion as delivered "astonishingly", General Campion is not singular in his fixation on seventeenth-century poetics (Ford 2012, p. 513). Numerous literary works of the 1920s quote Marvell's heroic couplets, and the aforementioned soldier protagonist of *Parade's End*, Christopher Tietjens, recurringly laments amidst the trenches, "What had become of the seventeenth century? And Herbert and Donne and Crashaw and Vaughan, the Silurist?" (Ford 2012, p. 611).

I would like to attempt to respond to this question. What has happened to the literary legacy of the seventeenth century, especially the metaphysical poets Tietjens lists, by the opening of late modernity? Why would one choose to mourn a dislocated, distant past amidst tangibly modern destruction? And what role could love poetry like Campion's favoured "To His Coy Mistress" have for a militaristic, imperial project? While countless modernist novels display an attraction to the power of early modern poetry amidst battle, *Parade's End* exhibits a particularly emphatic and extended focus on the relationship between poetry and war—and to an absurd degree, at least to an outsider. Soldiers "talk. . .in intimate undertones about the resemblances between the Petrarchan and the Shakespearean sonnet form" just before battle, they host timed sonnet competitions on Armistice Day and in the midst of fire (really!), and they recurrently quote (and misquote) poets like Marvell and George Herbert (Ford 2012, p. 643). Why is the love poetry of the seventeenth century relevant for examining amid modern conflict and its modernist literary coverage?

In his joint status as a formative modernist experimenter and volunteer soldier, Ford offers a particularly striking case study to examine the relationship between poetic tradition and imperial conflict. While enthusiast scholars have engaged in some analysis of his own poetry, his influence on later innovation, and his treatment of recent predecessors like Dante Gabriel Rossetti, there has been less sustained critical attention to his complex relationship to literary tradition, which is most vibrant, I would argue, in the characters' curious apotheosis of seventeenth-century poetics. Tietjens has often been considered a man out of time; his deep roots in the seventeenth century in particular have not been thoroughly unearthed. Meghan Marie Hammond defines Tietjens as an "eighteenth-century man[8] caught in the first great conflict of the twentieth century" in his promotion of the "eighteenth-century virtue" of "sympathy" (Hammond 2015, p. 65). Yet to fully grasp the heights of his strange, salvific vision that extends beyond "fellow feeling" to something perhaps even grander, this paper argues for the need to travel even further back in time, for in Tietjens' words, the seventeenth century might as well save one man!" (Ford 2012, p. 611). As Sara Haslam argues in another study of Tietjens' eighteenth-century influences, "There was, quite simply, more to history in *Parade's End* than the war", and I would further refine this statement: there is more to history in *Parade's End* than the eighteenth century (Haslam 2014, p. 38). Indeed, Haslam notes that "there is not space here to explore the theological aspects of Christopher's seventeenth-century allegiances here", "significant" though they are (Haslam 2014, p. 47). While Melanie East has engaged in helpful theological work in outlining the Christological aspects of Christopher Tietjens as a "Christ figure" who is "more confrontational than comforting", I seek to delve further into the seventeenth-century devotional poetics she mentions briefly (East 2023, p. 424).

In the ampler space of these pages, then, we will "go round and round" further back temporally, as I will outline the importance of seventeenth-century metaphysical poetry for modernist writers (especially but not limited to Ford) amid the context of muddy trench warfare of the First World War and analyse how *Parade's End* employs allusions to the poets Marvell and George Herbert to show early modern poetry's ambivalent capacities to fuel, escape, and critique modern warfare—to dwell, in short, in the ever-complicated tug of war between the aesthetic and the ethical. While high modernist writing traditionally prioritises the former over supposedly stilted Victorian moralisms, I argue that Ford utilises vital poetic allusion to attempt a thrilling, hopeful union of poetic justice through his revisions of early modern devotional poetics.

## 2. (Early) Modernism: "Heterogenous Ideas Yoked by Violence"

To better understand this modern(ist) temporal dialogue, let us "travel back" and look at the wider literary context of the early twentieth century before dropping any further into the trenches of Ford's particular corner of this cosmos. Writing on the emergence of European modernism in the battles of the First World War, poet Dana Gioia summarises: "For poets, the unprecedented scale of violence annihilated the classic traditions of war literature—individual heroism, military glory, and virtuous leadership. Writers struggled for a new idiom commensurate with their apocalyptic personal experience" (Gioia 2020, p. 4). But while poets and novelists give birth to radically new forms in this period, I'd suggest that this "idiom" has been curiously midwifed by early modern poetry. As I have mentioned, the characters of *Parade's End* do not have a monopoly on their obsession with the seventeenth century. In a puzzlingly funny moment amid the sad story of Septimus Smith in Virginia Woolf's *Mrs. Dalloway*, when we learn the strange reasoning behind his decision to fight in the First World War: "He was one of the first to volunteer. He went to France to save an England which consisted almost entirely of Shakespeare's plays and Miss Isabel Pole in a green dress walking in a square" (Woolf 1996, p. 64). Even ignoring the oddness of the green dress, his reasoning raises questions: why exactly have romantic love, early modern verse, and late modern English militarism been placed together? As the novel continues, Septimus' rambling post-war "writings" continue a backward *danse macabre* in describing "how the dead sing behind rhododendron bushes; odes to Time; conversations with Shakespeare" and "universal love" (Woolf 1996, p. 107).

It is unclear how all these obsessions connect, but lonely Septimus is not alone in his modernist, militaristic devotion to both women and early modern poetry amid infernal, temporal confusion. While surveying the soldiers under his command, *Parade's End*'s Tietjens stands (in a repeated comparison) "like Shakespeare contemplating the creation of, say, Cordelia" (Ford 2012, p. 559). This literary significance infused actual soldiers as well: in his landmark, polemical study, the 1975 *Great War and the Modern Memory*, Fussell punningly proclaims that "Oh what a literary war" the conflict was, as "there are some intersections of literature with life that we have taken too little notice of", like poet Siegfried Sassoon reading Thomas Hardy in a trench dugout (Fussell 2013, pp. 168, 8). More recently, Meredith Martin has argued that real-life "soldier-poets were conditioned to see themselves... as part of a collective English culture, bound to defend the language of Shakespeare" before and during the First World War, much like fictional Septimus (Martin 2012, p. 149). And soldier-poet Rupert Brooke, known to his contemporaries as "our Donne Redivivus", was inspired by the language of another early modern poet: for Joseph Bristow, his scholarly "admiration for Donne's resilient faith" emerges in his contested "Soldier"'s "English heaven" (Bristow 2014, pp. 679, 676).

As Brooke's English wording implies, soldier's literary language was often explicitly nation-based, and *The Great War and Modern Memory* has received critique for the unacknowledged "limits of Fussell's argument" in its "Anglo-centric" view (Jay Winter in the "Introduction" to Fussell 2013, p. xii). My study of this "literary war" will exclusively and explicitly focus on the affordances and limitations of an Anglo-centric imaginary in one particular English author in Ford.[9] However, I note that early English poetry's ability to capture the

ephemerality of life extends across the Atlantic (and into the fictional realm) in Ernest Hemingway's 1929 *A Farewell to Arms*, as protagonist Frederic Henry quotes Marvell's "To His Coy Mistress", to his consummated mistress before he must go back to war. After hearing, "but at my back, I always hear time's winged chariot hurrying near" the pregnant, soon-to-be-abandoned Catherine Barkley dryly identifies the poem and its divergence from her current state (Hemingway 1997, p. 143). T. S. Eliot's "Prufrock" also quotes (and tweaks) the poem to a romantic conquest, as he hopes "to have squeezed the universe into a ball/ To roll it towards some overwhelming question," but has not had as much success as Hemingway's hero (Eliot 1988, p. 60). So why this Early Modern(ist) fixation, for a movement famously focused on Ezra Pound's plea to "make it new", and especially in the wake of a war promising to usher in a new world order?

Much of T. S. Eliot's own prose helps explain and historically caused the modernist fixation on early modern poetry, so let us turn to his influential 1921 essay, "The Metaphysical Poets", for some guidance. Writing against the grain of longstanding tradition led by Samuel Johnson (in 1781) that considers "the race of writers that may be termed the metaphysical poets" to be "not successful", and even to "lose their right to the name of poets" in the "perverseness" of their expressions, Eliot celebrates these long-overlooked seventeenth-century poets such as Marvell and Herbert (whom we will later discuss) because they "possessed a mechanism of sensibility which could devour any kind of experience", perhaps even twentieth-century military discord (Johnson 1858, pp. 11–12; Eliot 2006, p. 198). Indeed, while Johnson complains of "the *discordia concors*" or "combination of dissimilar images" when "the most heterogeneous ideas are yoked by violence together", Eliot sees modernist value in a discordant mosaic of meaning (Johnson 1858, p. 12). While Eliot himself never saw combat, I contend that this "violent" metaphysical combination of "thought and feeling" was particularly compelling for soldiers attempting to come to emotional and logical grips with the horror of the First World War (Eliot 2006, p. 199). Neither the emotional distance of Victorian poets nor the logical justifications of classical war poetry could suffice for the modern experience of war. Instead of just sentimentally, insufficiently looking into the heart, Eliot shows how the "metaphysical" poets also "look into the cerebral cortex, the nervous system, and the digestive tracts", which was particularly fruitful for the psychological and physical toll of modern trauma like shell shock (Eliot 2006, p. 200). Indeed, for Martin, while "soldier-poets recognised their relationship to poetry and poetic form as equivocal, volatile, and distressing", the process of writing traditional poetry was a medically prescribed way of therapeutically processing trauma for soldiers like Wilfred Owen and Robert Graves, and "poetic meter was increasingly seen as a symbol of English national culture in this period" (Martin 2012, pp. 148, 147).

Quoting seventeenth-century authors in the midst of a global clash of waning empires could fantasise about a Europe devoid of modern weaponry and appeal to a glorified English past in its imperial heyday. Indeed, in her study of *Parade's End*'s temporal politics, Isabelle Brasme argues that the metaphysical poets represent "age-old resistance of insular values to continental invasion, a theme that was reactivated at the time of the First World War" (Brasme 2015, p. 3). By contrast, the advent of the eighteenth century and later Victorian "sentimental age" is marked by "the dissociation of sensibility" which Eliot argues dominated poetry after the metaphysical poets could not sufficiently adapt to the logical and emotional hurdles of the war, for Victorian poets "do not feel their thought as immediately as the odour of the rose" (Eliot 2006, p. 198). However, to misquote Septimus Smith's favourite near-metaphysical poet Shakespeare, do these violent metaphysical delights have violent ends? In other words, does poetic tradition catalyse conflict or help resolve it?

## 3. Setting the Scene of Battle

To help answer this question, I argue that Ford Madox Ford explores the political efficacy of metaphysical poetry most strikingly amidst the early modern fans amongst the modernists and in a way that prophetically anticipates contemporary academic treatment

of the mutually constitutive nature of war and sexual conquest. While Ford is perhaps best known for the pre-war ironic anti-heroism of the 1915 novel *The Good Solder*, the four novels comprising *Parade's End* chronicle the psychological experiences of British men and women before, during, and after the First World War with far richer political and poetic complexity (and much more confusion and heft). Developing upon its taut predecessor's impressionistic style, this epic tetralogy brims with disorienting modernist experimentation (in the disordered, fractured plot that represents shell shock), stirringly lyrical nostalgia, and futuristic, Christological hope.

This temporal tension is key: *Parade's End* is deeply invested in investigating Englishness, more specifically, for Brasme, the "*when* of Englishness", in order to understand and potentially reshape English identity and British imperial control (Brasme 2014, p. 1). The work's archetypal Englishman, protagonist Christopher Tietjens, volunteers for service in the First World War and depends on early modern poetic remembrance to get through the muddy, shell-shocked trauma of battle against both German soldiers and his wife Sylvia. Like his godfather General Campion (who incidentally pursues an affair with Sylvia), Tietjens possesses "a belief that all that was good in English literature ended with the seventeenth century", but the experimental tetralogy does not solely descend into melancholic nostalgia (Ford 2012, p. 394). While most of Ford's contemporary and more-canonised modernist writers like Hemingway and Eliot seem politically trapped in aestheticised melancholia[10], Paul Saint-Amour argues that through the novel's "untimely protagonist, one not the contemporary of his contemporaries; Ford will attempt to warn his own contemporaries away from war; through a futureless character...*Parade's End* tries to undoom the future of the world" (Saint-Amour 2015, p. 271). While Fussell's sweeping cultural history argues that this tragic war "reversed the Idea of Progress" in its "hideous embarrassment to the prevailing Meliorist myth which had dominated the public consciousness for a century", I contend Ford's futureless character offers an idiosyncratic reinterpretation of progressive ideas, perhaps without the capital "P" (Fussell 2013, p. 8).

We can see the seeds of the "undooming" Saint-Amour applauds alongside a hint at Fussell's sense of regress combined with Eliot's rejection of Victorian sensibility early in the pages of *Parade's End*, and its turn to the metaphysical poetry Eliot celebrates takes centre stage as the plot and the war develops. At the start of the first novel set in 1912, Tietjens' friend, a Victorianist Dante Gabriel Rossetti scholar named Vincent MacMaster, insists that "a war is impossible", but Tietjens disagrees and correctly predicts, in a fascinating theorisation of the roots of conflict combining sexual and imperial conquest:

> Yes, a war is inevitable. Firstly, there's you fellows who can't be trusted. And then there's the multitude who mean to have bathrooms and white enamel. Millions of them; all over the world. Not merely here. And there aren't enough bathrooms and white enamel in the world to go round. It's like you polygamists with women. There aren't enough women in the world to go round to satisfy your insatiable appetites. And there aren't enough men in the world to give each woman one. And most women want several. So you have divorce cases. I suppose you won't say that because you're so circumspect and right there shall be no more divorce? Well, war is as inevitable as divorce... (Ford 2012, p. 22)

In the dismissive "you fellows", Tietjens decries a certain sentimental type of poetic temperament. The narrator remarks that he is "jibing again at the subject of Macmaster's monograph", the Victorian poet (and loose relation of Ford's[11]) Dante Gabriel Rossetti (Ford 2012, p. 22). His like-minded love and ultimate mistress Valentine Wannop, a "New Woman" figure (whom Tietjens first encounters while she is blowing up his golf course in a suffragette demonstration and with whom he falls in love while she corrects his Latin poetry errors), agrees—she mocks Macmaster's mistress' pretentious "sets of quotations for appropriate occasions": Dante Gabriel "Rossetti for Love; Browning for optimism" amid other "shallow" pieces of Victoriana (Ford 2012, p. 548). Ultimately, as Alec Marsh has written, "the titillating 'aesthetes' of the later nineteenth century [are] replaced by deeper seventeenth-century voices" and "the shaky morality of Rossetti[12] [is] exchanged for the

upright virtue of George Herbert" (Marsh 2014, p. 167). Well, not quite "upright". In choosing to pursue sexual communion outside of marriage, Tietjens' own mistress mourns that her beloved can never be a "country parson" like Herbert, but he can perhaps pursue a new form of sainthood in innovative conversation with the past virtue. Dialogue with these seventeenth-century voices does not arrive until the trenches, but this early pre-war passage sets up the novel's futuristic vision of the complex relationship between imperial expansion, financial greed, and sexual conquest. And, of course, poetic tradition.

But before we study how it is allusively articulated, what guides this "undoomed" future in its unlikely combination of saintly verse and sexual sin? Some form of Christology, for one: this untimely protagonist, volunteer soldier "Christopher" Tietjens is a Christ-bearer, etymologically, at least, and he shows a new way, or *when*, of thinking in his status as a sacrificial soldier fighting for an earlier era to promote progress—not Benjamin's "homogenous, empty time of modernity" but something perhaps approaching his mystic "messianic time" (Benjamin 2019, p. 264). In the philosophical structure of the singular "now" of a "monad" that allows a departure from temporal homogeneity, Benjamin "recognizes the sign of a Messianic cessation of happening, or put differently, a revolutionary chance in the fight for the oppressed past", and through the monadic figure of Tietjens, the reader encounters a strange, backward-glancing yet revolutionary hero (Benjamin 2019, p. 257). In memories of car crashes, mutilated bodies, and a deeply marred marriage, like Benjamin's "Angel of History", Tietjens' "face is turned toward the past" and "sees one single catastrophe which keeps piling wreckage upon wreckage", but he nevertheless finds something to hope for (Benjamin 2019, p. 257). For in Ford's strange soteriology,[13] I would argue that we see not a "chain of events" but a catastrophic apocalypse of nonlinear writing in which "every second" is "shot through with chips of Messianic time" (Benjamin 2019, pp. 263–64).

What new heaven or earth does the angel see in our modernist, backward-looking Christ's resurrection? Not an entirely conventional one. In rejecting British emotional suppression and breaking his marriage vows to pursue his true love and intellectual equal in Valentine as his mistress, he articulates a strange new vision of loving Christian communion of interpersonal transcendence in a fascinating (a)sexual fantasy amid the muddy trenches. Amid the aforementioned backdrop of Marvellian seduction poetry, Christopher offers the unintuitive reformulation of modern love:

> You seduced a young woman in order to be able to finish your talks with her. You could not do that without living with her. You could not live with her without seducing her; but that was the by-product. The point is that you can't otherwise talk. You can't finish talks at street corners; in museums; even in drawing-rooms. You mayn't be in the mood when she is in the mood--for the intimate conversation that means the final communion of your souls. You have to wait together--for a week, for a year, for a lifetime, before the final intimate conversation may be attained...and exhausted. So that...

> That in effect was love. It struck him as astonishing. (Ford 2012, p. 629)

Christ bore his own astonishing vision first to women at the tomb, and Christopher's vision of atypical modernist love is a relatively feminist one—Tietjens suggests that "the future of the world [is] to women", and in a strange sort of erotic conversion narrative, he chooses to leave his society wife Sylvia for the company and politics of the pacifist suffragette Valentine (Ford 2012, p. 629). This feminist vision is not limited to Christopher: in her exploration of the tetralogy's "Articulations of Femininity", Brasme summarises that Ford's "feminism surfaces...through the way in which he allows his narrative to be constantly inhabited by women's voices" (Brasme 2014, p. 184). Indeed, though Ford himself had complicated relationships with the opposite sex (first mentoring then romantically jilting Jean Rhys[14]), he was an avowed suffragette. While Tietjens' amatory dogma is seemingly similar to Macmaster's exaltation of sex, this picture really functions as its opposite: sex as object and seducee as objectified are replaced here by transcendent, egalitarian love by means of sex.

This end of sexual oppression and of the entrenched battle of the sexes is deeply imbricated, as we shall see, with literal battle, in Ford's (and heroine Valentine's) overarching, explicitly stated purpose of pacifism. While he began the war working for the War Propaganda Bureau, as Brasme outlines, Ford's stances deepened as the fighting progressed with regards to "the ethical dimension of literary creation" universally, in his "feeling of duty toward humanity as a whole" (Brasme 2020, p. 7). His stances had dramatically and explicitly shifted by the 1920s; in his memoir, Ford notes that while he previously had "the greatest contempt for novels written with a purpose", *Parade's End* was intended as a work that "should have as its purpose the obviating of all future wars" (Ford 1933, p. 205).

The pacifistic, feminist future this novel attempts to imagine should be fairly clear. But how exactly do we achieve this vision through a backwards protagonist with a "belief that all that was good in literature ended in the seventeenth century" (Ford 2012, p. 394)? This article attempts to reconcile this temporal fissure by arguing that the devotional aspect of early modern poetry has a very complex but powerful role in shaping Ford's political purpose in this strange conversion narrative. Contemporary peace and conflict studies scholars such as Holly Furneaux situate their literary criticism "within continuing, broader scholarship about the relationship between emotion, war, and the possible end to war" with the hope of the "potential to change tolerances for war", and while Furneaux focuses on the earlier Crimean War, and to trace the pacifistic uses of early modern poetry in the modernist vision, I will extend her refreshingly idealistic study in both directions (Furneaux 2016).

## 4. (Early) Modern Love

For Tietjens' constellation of complex causes of "inevitable" war is not distinct to late modernity: sex, war, and poetry have been triangulated at least as far back as the *Iliad*, and it particularly evokes an earlier age of England, such as seventeenth-century poetics scholar Roland Greene's depiction of "the mutuality of love and empire, and its implications for how we read early modern literature" in his classic *Unrequited Conquest* (Greene 1999, p. 33). Tietjens' beloved Donne compares his "Mistress Going to Bed" to an "empirie" and "all states", and an understanding of the imbricated nature of sexual and imperial conquest has great implications for how we read how *modernists* like Ford read early modern literature as well. We can see the modern(ist) inheritance of Roland Greene's unified discourse of love and empire in the muddled scene with which this essay opened, for General Campion's allusion to Marvell in the second novel of the tetralogy clearly displays "the interpersonal, the social, the political, and the religious senses of love may animate each other" (Greene 1999, p. 23). Hundreds of pages and several years have passed since the beginning of the story's pre-war, idyllic, Rossetian days, and "no more parades" of sentimental, honourable Victoriana are in sight, as the second novel's title contends. In order to describe the existential despair of his "general's life in this accursed war" Campion quotes:

But at my back I always hear

Time's winged chariot hurrying near:

And yonder all before me lie

Deserts of vast eternity! (Ford 2012, p. 511)

While Marvell's original verses argue that life is tantalisingly short in order to seduce his "coy mistress", and Hemingway's Frederic Henry quotes it to a woman he has already seduced, Campion quotes these lines to another soldier, seemingly to indicate a renewed wartime horror at the ephemerality of life and finality of death amid shell bombardment. He continues to selectively quote the poem's depiction of death later in the conversation. As he notes:

But there are finer things in Marvell than that. . .

There's, for instance:

'The grave's a fine and secret place

But none I think do there embrace. . .' (Ford 2012, p. 512)

Interestingly, Campion misquotes the poem in both of these instances, changing "yonder all before *us* lie" to "before *me lie*" and "the grave's a fine and private place" to "secret place", which perhaps indicates the temporal fissures between early modern and modernist society—here we may thus glimpse an increased sense of isolated individuality. Indeed, there is no "us" of seduction here, just individual soldiers' existential despair and imprecision. T. S. Eliot quotes Marvell for Prufrock's attempt to woo a woman, and after hearing about a soldier who tries and fails to work up the courage to get divorced after his wife seduces him, Campion proclaims, "*that's* modernism" amid the sexual yearning of seduction poetry (Ford 2012, p. 529). Divorce was much less attainable in the seventeenth century, of course, but its overt sexual desire appeals to a wasteland of death; Eros and Thanatos coalesce in early modern and modernist love. While Woolf's Clarissa Dalloway seems to respond to Marvell in her final revelation of the distanced, sublimated erotics of "an embrace in death" when she hears of a stranger's suicide, *Parade's End* insists on real seduction amidst the infernal trenches (Woolf 1996, p. 134).

For there is more to the ideological dimension of seduction poetics than first meets the eye—as we later learn, General Campion is currently engaged in winning over his godson Tietjens' wife Sylvia. Sylvia herself notes candidly a few pages before: "This whole war was an agapemone. . . You went to war when you desired to rape innumerable women. . .It was what war was for. . .All these men, crowded in this narrow space. . ." (Ford 2012, p. 428). Again, Ford anticipates Greene's early modern theorisations. Sexual and imperial exploitation are mutually, fatally allied—a time-transcending truth, perhaps, but rendered more corrosive in the brusqueness of modernist prose and the seams cracking in the late-stage British empire. While perhaps more explicit in Ford's raw twentieth-century language, looking further into the love discourse of Marvell's poem may reveal the roots of English imperial love:

> We would sit down, and think which way
>
> To walk, and pass our long love's day.
>
> Thou by the Indian Ganges' side
>
> Shouldst rubies find; I by the tide
>
> Of Humber would complain. I would
>
> Love you ten years before the flood,
>
> And you should, if you please, refuse
>
> *Till the conversion of the Jews.*
>
> My vegetable love should grow
>
> *Vaster than empires and more slow* [. . .] (Marvell 1681, pp. 19–20 emphasis mine)

Like Greene's unified discourse of imperial lust, Marvell's love is measured in terms of empire expansion and in rubies gained. Here, the poetic speaker links the desire for riches, sexual consummation, and imperial conquest. While antithetical to Eliot's ahistorical, New Critical assessment of metaphysical poets like Marvell, it has become more or less a critical commonplace to link Marvell's love poetry with imperial desire and his position as a member of Parliament. For example, Michael McKeon argues that "Marvell's poetry, whether political or pastoral, amatory or devotional, is deeply concerned with the imperialistic and deeply problematic relation between unequal entities" in his portrayal of "imperious loves" (McKeon 1983, p. 46). For McKeon, almost all his poetry is "bound together by the seventeenth-century notion and language of empire", such as Marvell's celebration of Puritan colonisation of Edenic "Bermudas" (McKeon 1983, p. 46). In the seventeenth century, Britain's imperial hold on India was still in its initial supposed "trading" phases, but by the early twentieth century, its dominance was at its height. And in an eerily fitting textual congruity, at the close of *Parade's End*, Campion has successfully seduced Christopher Tietjens' wife, and the two of them are heading to occupy his post as the Viceroy of India. Pleasing rhymed couplets from England's early glory days make this conclusion all the more palatable, at least for Campion. The novel does not necessarily sanction these imperial events, but it does show the power of a certain valence of poetry to catalyse romantic and political dominance.

### 5. Pacifist Poetics

While we see here that early modern poetic tradition can be used to successfully fuel imperial and sexual conquest in the case of Campion, *Parade's End* shows at greater length with its protagonist that it can wield the opposite effect, especially in this poetry's devotional capacity. Saint-Amour remarks that "in its great goal of obviating all future wars, *Parade's End* might have found a powerful instrument" in Campion's "ability to connect military and private life, war and empire", somewhat like Greene (Saint-Amour 2015, p. 287). But while the figure of Campion unpacks the rhetoric of imperial lust, Ford ultimately "routes his anti-war polemic" through "the minute-by-minute psychological experience of the soldier" Tietjens, and his intense fixation on a more psychologically insular, devotional metaphysical poet (Saint-Amour 2015, p. 287). Shortly after Campion quotes Marvell, Tietjens begins to fantasise about Welsh-born priest George Herbert:

> Tietjens was sentimental at rest, still with wet eyes. He was walking near Salisbury in a grove, regarding long pastures and ploughlands running to dark, high elms from which, embowered...Embowered was the word!—peeped the spire of George Herbert's church... One ought to be a seventeenth-century parson at the time of the renaissance of Anglican saintliness... who wrote, perhaps poems". (Ford 2012, p. 535)

After this sentimental backward glance of a reverie ends, Tietjens summarises, "That was home-sickness! ...He himself was never to go home!" (Ford 2012, p. 535). The second novel of the tetralogy ends in this nostalgic despair for a seventeenth-century British fantasy that cannot match his horrific present in the trenches of France. But in the next volume of the work, Tietjens' idealisation of Herbert moves from a passing fancy to an imperial rejection. He dreams of Herbert's Bemerton parsonage, an idealised space and time outside of the thorny warring present, and uses it not only to escape but also to question his current political situation.

In the third and more optimistic novel of *Parade's End*, *A Man Could Stand Up–*, Tietjens repeatedly quotes Herbert's "Vertue" (from his 1633 The Temple) for solace for and beyond his individual soul (he mainly quotes the first line, but I will include some of the rest of the poem for context):

> Sweet day, so cool, so calm, so bright,
>
> The bridal of the earth and sky;
>
> The dew shall weep thy fall to-night,
>
> For thou must die.
>
>         [. . .]
>
> Only a sweet and virtuous soul,
>
> Like season'd timber, never gives;
>
> But though the whole world turn to coal,
>
> Then chiefly lives. (Herbert 1903b, p. 80)

Fussell notes that "Christopher Tietjens' obsession with 'Virtue'" among other examples, displays how "the *Oxford Book of English Verse* presides over the Great War in a way that has never been sufficiently appreciated", and I would like to appreciate the significance of this example in greater depth than afforded in Fussell's list (Fussell 2013, p. 172). Like Marvell, Herbert's poem speaks to that age-old human concern that was especially trenchant in the trenches of the First World War—how must we act when starkly confronted with our mortality? For Marvell, it is sexual (and latently imperial) conquest in what time we have. For Herbert here, the solution is to turn inward—to cling to oneself and, more importantly, to one's God. In a telling contrast that implies the great distance within the classic "metaphysical" grouping, Marvell asks us to seize the day; Herbert asks us to weep for its death and hope for its eternal resurrection. Herbert's poem grants Tietjens immense spiritual solace in the godless warfare of WWI and offers a catalysing point for reimagining,

or rather re-temporalizing, English politics. He also quotes Herbert's "The Elixir" as "the equivalent of prayer" before battle, as Haslam's notes to *A Man Could Stand Up*—highlight (Ford 2011, p. 126). In a subtle but telling difference, Tietjens changes God's "laws" to a higher "cause" in quoting the line "who sweeps the room as for thy cause" for leaderly inspiration, which perhaps indicates Tietjens' preference for a spirituality not based on conventional "rules" that would deny him the chance for the higher love of his desired mistress Valentine (Ford 2012, pp. 125–26).

As the novel progresses, Tietjens continues to distinguish his preference for seventeenth-century insular, pastoral spirituality over the Elizabethan "preposterous drumbeating" a figure like Campion perhaps inherits:

> Heaven knew, we did not want a preposterous drumbeating such as the Elizabethans produced—and received. Like lions at a fair… But what chance had quiet fields, Anglican sainthood, accuracy of thought, heavy-leaved, timbered hedgerows, slowly creeping plough-lands moving up the slopes? …Still, the land remains… The land remains… It remains! …At that same moment the dawn was wetly revealing; over there in George Herbert's parish… What was it called? …What the devil was its name? Oh, Hell! …Between Salisbury and Wilton… (Ford 2012, p. 612).

For Tietjens, Herbert's metaphysical epoch is "the only satisfactory age in England… yet what chance had it today? Or, still more, to-morrow?" (Ford 2012, p. 612). Initially, the answer seems none, no chance—reclaiming George Herbert's pastoral ideal in the muddied trenches of a different century seems fantastically unattainable, especially because Tietjens cannot even remember the name of it (due to shell-shock-induced memory problems). But all is not consigned to half-remembered oblivion—Tietjens does remember the name of the parish (Bemerton) after sustained mental exercise, and his poetic allusion builds to an imperial critique. Describing Herbert's parish, he thinks:

> What a handful of frail grass with which to stop an aperture in the dam of— of the Empire! Damn the Empire! It was England! It was Bemerton Parsonage that mattered! What did we want with an Empire? It was only a jerry-building Jew like Disraeli that could have provided us with that jerry-built name! The Tories said they had to have someone to do their dirty work… Well, they'd had it! (Ford 2012, p. 639)

"Dam" transforms into damnation to obstruct imperial flow in this radicallyrooted stream of consciousness. In Tietjens' mind, the sacred particularity of Bemerton Parsonage's dewy grass can somehow replace the seductive claims to universality of the British Empire. Indeed, Tietjens continues to not just imagine Herbert's words but the particular terrain they are rooted in, as he ecstatically imagines the sun "rising on Bemerton!" in a hopeful backdrop to his repetition of "sweet day so cool, so calm, so bright (Ford 2012, p. 634). While Fussell, using the (brief, uncontextualised) example of Tietjens, describes the "cruel reversal that sunrise and sunset, established by over a century of Romantic poetry and painting as the tokens of hope and peace and rural charm, should now be exactly the moments of heightened ritual anxiety" I argue that Ford's use of devotional poetry *does* retain an explicitly sunny hope against hope, not just ironic, tragic, anxiety (Fussell 2013, p. 56). Even in Ford's nonfiction, Herbert's verse along with his "vision of a perfected High Church Earth" at near Salisbury marks the "final height to which devotional poetry may attain", and its invocation helps Tietjens attain a political vision of the prince of peace (Ford 1994, p. 488). While Campion employs Marvell to establish his conquesting desires, Tietjens employs Herbert to reimagine and ultimately reject the imperial trenches of the present through an excavation of sedimented literary layers which can uncover an earlier, more insular version of England where "a man can stand up," like Herbert at Bemerton, "upon a hill" (Ford 2012, p. 656).

### 6. The Implications of Isolationism

But pastoral as these poetics may be, this anti-imperial fantasy is not without its complications. The contrast between Herbert and Marvell's usage shows that metaphysical poetry can be used for quite different purposes, but the shared appeal to a bygone era has its ideological overlaps. Racial and religious purity and English supremacy can accompany imperial as well as insular "Little England" politics. Indeed, Tietjens (or is it Ford?) actually mixes up his metaphysical poets and mistakenly thinks that Campion earlier quoted Herbert, which may subtly show the potential slippage between these two ostensibly different poetic purposes, amatory and devotional (which we can also see in the sacred erotics of another metaphysical poet, John Donne). In his confused recollection, Tietjens' consciousness streams:

> Sweet day so cool, so calm, so bright, the bridal of the earth and sky! . . .By Jove, it was that!. . .Old Campion, flashing like a popinjay in the scarlet and gilt of the Major-General, had quoted that in the base camp, years ago. Or was it months? Or wasn't it: 'But at my back I always hear Time's winged chariots hurrying near', that he had quoted? (Ford 2012, p. 611)

Tietjens does not just mess up the poets—he slightly changes the poetry, as Haslam's notes to the volume indicate that "'chariot'" is singular" in the original, and his remediation of the past may lose some legitimacy when he has no confident sense of his own time passing (Ford 2011, p. 88). Misquotation may seem minor, but it can be telling: Marvell's speaker temporally connects the seduction of his beloved to the "conversion of the Jews", and Tietjens similarly builds its pastoral, anti-imperial fantasy in contrast with the "jerry-building Jew Disraeli" (Ford 2012, p. 639). He blames the empire on the nineteenth-century Jewish Prime Minister Benjamin Disraeli, but the roots of empire can be clearly found in the early modern period he idealises.

Indeed, while the Herbert ensconced in his Bemerton parsonage was not as directly connected to imperial development as Parliamentarian Andrew Marvell, Herbert was quite ideologically committed to transatlantic expansion. As John Kuhn has argued in his re-contextualization of Herbert's writings such as "The Church Militant", "Herbert's desire to work in 'New Jerusalem' was anything but figurative" (Kuhn 2016, p. 65). This idealised New Jerusalem is not just a state of spiritual immortality achieved after a life of "virtue", but rather "a practical goal. . .fueled by the widespread suggestion that the New World of the English colonies might eventually be the site of the new world of the godly" (Kuhn 2016, pp. 65–66). It is outside of the scope of this paper to dwell in the minutiae of Herbert's religiously-infused politics (which seem markedly less antisemitic than Tietjens'[15]), but looking further into early modern scholarship displays that the completely insular rural England Tietjens idealises is to a considerable extent a constructed fantasy. As Saint-Amour remarks, "The condemnation of the imperial present, the idealisation of the rural past, the anti-Semitism alongside Anglo-Saxon 'cradle of the race' paeans to the Salisbury countryside—this is none other than. . . Little Englandism" (Saint-Amour 2015, p. 299). Through this nativist, alien-averse political orientation, Tietjens "assuag[es] the loss of British universalism through an embrace of English particularism" which is "nostalgic in its temporality" (Saint-Amour 2015, p. 299). The impending loss of empire is replaced by prejudiced isolationism, which is not necessarily a healthier political paradigm.

Of course, neither Herbert's views nor even Tietjens' are necessarily the novel's. Sara Haslam's notes to the critical edition of this volume that as late as 1914, in an *Outlook* essay, Ford quoted this Herbert poem as "a symbol of the distance he feels from his friend's 'High Church' religion" (Ford 2011, p. 87). Perhaps Ford's hellish time in the trenches brought him closer to a theologically-laden Herbert fantasy, but this does not necessarily rope him to conservatively nostalgic politics. As Max Saunders has shown in his study of Ford and race, throughout his life, "Ford was categorical and clear in his anti-imperialism," such as in his "support. . .for African and Irish home rule", and that he clearly "reproves the Little Englanders who want to keep out the alien" for "they are likely to be the product of something as 'other', or alien, as anything else" (Saunders 2011, p. 48).

Writing in the introduction of his critical edition of the first volume of *Parade's End, Some Do Not...* Saunders summarises that "Ford sees Englishness as the product of sustained immigration" (Ford 2010, p. xxxvii). Within the novel, the ideal of English racial purity seems completely untenable in practice with the relatively ethnically diverse characters of *Parade's End*—Christopher Tietjens himself is of Dutch origin (as his oft-mocked name implies), he ultimately goes into business with an American Jew, and Ford's own father was German.[16] And elsewhere, the novel strongly critiques antisemitism in its portrayal of the Dreyfuss affair. Moreover, the tetralogy's own experimental anti-linear form itself shows the importance of thinking beyond outdated aesthetic models, even as we may use them for (limited) inspiration.

### 7. Coda

For all its future-oriented modernism, then *Parade's End* displays early modern poetry's lasting power even as it articulates a new vision of transcendent community. Christopher Tietjens does not quite achieve the "Anglican sainthood" of George Herbert; in the final novel, he retires to the English countryside and keeps fantasising about Herbert, yes, to "buy a living where George Herbert had been parson" for his son, but to live quite modernly and seize the day with his mistress Valentine out of wedlock (Ford 2012, p. 882). The parade of artificial and often violent tradition ends, and Ford does not advocate for poetically infatuated time travel backwards, no matter how appealing it may seem. For Brasme, even as these novels appear to mourn for the passing of an English greatness, they "invite us to consider anew the pretence and fragility, if not hypocrisy, of national identity" as well as masculine bravado (Brasme 2015, p. 21). We see no simple Edwardian exaltation of "hearts at peace, under an English heaven" as Rupert Brooke's "Soldier" yearns, but a rearticulation of the universal in the intimate as articulated by a humble Welsh-born priest. Heaven may be rooted in the local, but Ford's use of Herbert's words orients us away from landed conquest to the transcendent. Empire be damned, interpersonal communion (whether in devotion to women, a poetic predecessor, or God) be saved. And unlike the fragility of intimacy in other aforementioned modernist writers—*A Farewell to Arms'* fatal childbirth, Septimus' suicide, and Prufrock's unanswered "love song"— in contrast to the dominant "tragic metanarrative" that Leonard V. Smith finds in many works about the First World War like Fussel's, *Parade's End* dares to imagine a hopeful postwar future full of (ir)reverent, continued consummation and new life (Smith 2001, p. 258).

Utopia on this earth is not so easy, of course, as the interpersonal and financial tensions of the last post-war volume of the novel display in its fraught pastoral aspirations and tensions. Ultimately, the complexities of the long-negotiated metaphysical allusions of this text help Ford to show us to be mindful of the nuanced, nationalistic ways poetry can sanction both imperial exaltation and damnation. Devotional verse may direct us to a complicated heaven, while unchecked amatory verse can lead us to imperial hell. In Tietjens' devotional and amatory finale that lives out his ideal of "communion of souls", Ford offers us a new vision of both masculine seduction and empire through dialogue for and against metaphysical poetry in a modern conversion narrative to feminist, pacifist devotional poetics. Unlike Eliot's essay, this multivalent work does not prescribe precisely how we should utilise poetic history, but Ford's temporal jumble encourages readers to take nuanced stock of our cultural memory and the ways it may help us reify or reimagine the failures of the present. And to write our own poetry, in dialogue or argument with history. For if poetry does anything, it keeps us living meaningfully in the face of impending death, like the soldiers writing sonnets in the trenches. I opened with some of my own attempts at (early) modernist poetics, and I'll close with some of Ford's own war verse, in which the past can be made anew:

I should like to imagine

A moonlight in which the machine-guns of trouble

Will be silent. . .

> Do you remember, my dear
>
> Long ago, on the cliffs, in the moonlight,
>
> Looking over to Flatholme
>
> We sat… Long ago!…
>
> We shall do it again
>
> (Ford 1971, p. 79)

Perhaps in this paper, I am a version of another MacMaster with an overinflated sense of poetry's import[17], or an excessively nostalgic Tietjens, or a Ford confusing his reader with an antilinear piece of writing. But like Benjamin, I have a metaphysical feeling that the backward glance can be a progressive force. For the moment, though, craning my head back, I see "time's chariot hurrying near", and cannot make the sun (or this paper) run any longer.

**Funding:** This research received no external funding.

**Data Availability Statement:** No new data were created or analyzed in this study. Data sharing is not applicable to this article.

**Acknowledgments:** Many thanks go to Jenna Lay and the members of her 2021 Early Modern Poetics seminar at Lehigh University, Charles LaPorte and Francesca Colonnese at the University of Washington, the editors of this special issue, and the anonymous peer reviewers for *Humanities* for a great deal of helpful feedback and encouragement on this article.

**Conflicts of Interest:** The author declares no conflict of interest.

## Notes

1. This poetic collage of an epigraph features lines from four poems spliced together from seventeenth-century poems associated with the "metaphysical school" later described in the essay. Lines one and two are from "The Retreat" by Henry Vaughan (1621–1695), a Welsh poet whose lost presence the protagonist briefly laments (Vaughan 1914, p. 419).

2. Lines three and four are from English poet Andrew Marvell's (1621–1678) seduction poem "To His Coy Mistress", which will feature prominently in the essay as it plays a major part in *Parade's End* (Marvell 1681, pp. 19–20).

3. Line five is drawn from Welsh-born English poet and parson George Herbert's (1593–1633) "Time" (Herbert 1903a, p. 124). This particular poem does not feature in the tetralogy (or this essay), but Herbert plays a major role, as does the fraught exploration of "time".

4. Finally, the final line six comes from Richard Crashaw's (1612–1649) "Caritas Nimia", or "The Dear Bargain" (Crashaw 1836, pp. 334–35). Crashaw was an English poet and cleric who converted to Roman Catholicism, and like Vaughan features briefly in the protagonist's lament of a bygone time.

5. The saga is composed of four novels: *Some Do Not…* (1924), *No More Parades* (1925), *A Man Could Stand Up*—(1926), and *Last Post* (1928).

6. Quite literally, my computer cannot seem to count the number of ellipses in the first novel alone—at least 1000.

7. All citations of *Parade's End* will be from the omnibus Vintage edition unless otherwise specified (occasionally, I cite footnotes from Carcanet critical editions). Additioanlly, all parenthetical citations refer to page number.

8. Multiple women in the novel describe him in this temporal way, including the leading ladies we shall soon discuss, his pacifist mistress, Valentine Wannop and his society wife, Sylvia.

9. Others have critiqued Fussell's unacknowledged bias toward "high culture", as Leonard V. Smith describes in his 2001 article "Paul Fussell's *The Great War and Modern Memory*: Twenty-Five Years Later" (Smith 2001, p. 243). Smith shows how Fussell gestures toward archival work on common soldiers but never takes up this work at length in his focus on "great" writers. I cannot claim to either Fussell's veteran experience or his archival work, though this essay was in part inspired by visiting the Imperial War Museum in London, whose materials Fussell cites. However, I can try to be more specific about the limitations of the vision of a single English author who is immersed in high culture. While Ford's work can be a fascinating starting point for rethinking how cultural memory functions in war and his literary predilections resonate with other soldiers to some extent, this one work by Ford cannot come close to speaking for "great war and modern memory" at large.

10. See Seth Moglen's (2007) *Mourning Modernity* for a fascinating account of the ways conservative critics canonised the supposed sophistication of politically "resigned" works and left out more radical responses to modern paradigm shifts. While he focuses on American authors, I would argue they cannot be isolated from the transatlantic culture of modernism and that Ford falls in the latter category.

[11]     Ford's mother's half-sister was married to Dante Gabriel Rossetti's brother, William Michael Rossetti.

[12]     The poetry of Dante Gabriel Rossetti's (then) less famous sister Christina (who Ford considered one of the best poets of the nineteenth century) receives better treatment in the novel, but the encyclopedia-minded Tietjens cannot remember it after the shell shock of battle (Ford 2012, p. 175). His vision of history must be strengthened by something further back than the short-term Victorian past.

[13]     I have not the space to detail Ford's own religious proclivities, but he was frustrated with contemporary religious ennui and converted to Catholicism at the age of 18, though his series of sexual affairs does not necessarily rhyme with traditional Catholic morality. For more on Ford's complex relationship with faith as "not a very good Catholic", see Max Saunders's biographies, especially his recent "critical life", *Ford Madox Ford* (Saunders 2023, p. 168).

[14]     For far more depth on these relational complexities, see Joseph Weisenfarth's (2005) *Ford Madox Ford and the Regiment of Women: Violet Hunt, Jean Rhys, Stella Bowen, Janice Biala* and his 2016 (Weisenfarth 2016) "'Quartet' with Variations: Ford Madox Ford, Stella Bowen, Jean Rhys, Jean Lenglet" (*International Ford Madox Ford Studies*). This paper will largely stay confined to the politics on the page and between the pages.

[15]     As Kuhn argues, "Throughout his works, Herbert urges his readers to identify with the figure of the Jew as the exemplar of the experience of divine abandonment, explicitly inviting the implied English (or colonial) reader to understand the Jewish experience as a version of the church's immediate future" (Kuhn 2016, p. 74). While Tietjens juxtaposes Englishness with Jewishness, Herbert perhaps advocates for uniting the two.

[16]     Ford's birth name was Ford Hermann Hueffer, and as Saunders quotes, Ford wrote in 1911 that "because I have a German name, I am often taken for a Jew," though he was a Catholic (qtd. In Saunders 2011, p. 47). Analysing Ford's complex personal relationship to English race and religion is outside the scope of this argument, but for more information, see Saunders' "Ford Madox Ford, Race and Europe".

[17]     While my self-analysis and use of the "I" may be untraditional, if not even more self-aggrandising, I do think Ford invites scholars to think metacognitively about the use and abuse, power and glory of our own words.

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
