# Peer review of "“Damn the Empire!”: Imperial Excess, National Nostalgia, and Metaphysical Modernism in the Poetics of Parade’s End"

_humanities, doi:10.3390/h13020065_

Round 1
Reviewer 1 Report
Comments and Suggestions for Authors
This is a lively, engaging, and - I believe - original paper that reveals the depth of Ford Madox Ford's engagement with 17th-century literature as articulated through the Parade's End tetralogy. It brings two cultural moments into direct conversation and argues neatly that the kind of imperial bombast associated with WWI is at turns encouraged and critiqued by writers from the earlier period. A similar dialectic is set up in the Fordian milieu, whereby the imperialist sentiment that fuelled the war can also turn to a kind of isolationism that eschews all kinds of contact - including conquest - with the outside world. The tensions and connections between sexual conquest vs. egalitarian sexuality, and imperial might vs. spirituality and pacifism. are elegantly drawn out. The author is clearly well informed on both canonical and lesser-known 17th-century writers. I particularly appreciated the creative playfulness of beginning the paper with a bricolage of verse.
I do not consider that it requires any revision beyond a few typos/phrasing issues, but there may be an opportunity to include more contemporary literary context when concluding the article. this could circle back to useful observations and arguments made at ll. 53, 62. 143-4, and 375. The nod to Benjamin is very welcome and perhaps inclusion of the 'monad' idea would work well in this paper that in effect fuses two 'moments of danger'.
I would draw the author's attentions to the following line numbers to check for style/typographic errors:
74, 77, 86, 91, 96 (is 'admits' right?), 101 ('only' is superfluous), 121, 165, 184 (query 'imagine'), 200-1 ('much more overlooked' is infelicitous), 204, 208 (no itals for tetralogy name), 249 (correct to 'with whom'), 259, 274 (suggest 'an entirely'), 290 ('fairly' and 'relatively' in same phrase seems excessive), 320, 337-340 (difficult to follow), 375, 419 ('wield' rather than 'possess'?), 455, 558, 572.
The following comments follow the prompts suggested by the journal.
-
Novelty: Is the question original and well-defined? Do the results provide an advancement of the current knowledge?
Yes. The argument is novel and will benefit Ford scholars, 20th c literature and history scholars, Early Modernists, and war poetry academics.
-
Scope: Does the work fit the journal scope*?
A perfect submission for this special issue. -
Significance: Are the results interpreted appropriately? Are they significant? Are all conclusions justified and supported by the results? Are hypotheses carefully identified as such?
Yes -
Quality: Is the article written in an appropriate way?
Yes, with pleasing whimsy. I did feel it got a bit too 'Dear Reader' at the close and consider it would conclude more emphatically at l.581; but this is a question of style not content.
Author Response
Thank you for the encouragement and suggestions; in the attached file I have described how I've incorporated your advice.

Reviewer 2 Report
Comments and Suggestions for Authors
This submission’s second half contains strong close readings of the intertextuality of Ford’s Parade’s End and the metaphysical poets the novel quotes and references. At its best, like the section on Marvell (lines 376-415), it makes a compelling argument for the living presence of 17th century poetry in the tetralogy, and the author here displays a grasp of the critical debates surrounding both early modern and modernist literature. Subsequent sections dealing with Herbert also succeed in showing the reader what happens when we take Ford’s poetic allusions seriously. My only suggestion for this part is to go back to Parade’s End, and especially to the annotated editions that highlight literary allusions, and bring in as many as you can to show the reader just how engaged Ford is with the 17th century. The author does not have to do in-depth like they do in the existing sections, but just convince the reader that those deep moments are motivated by Ford’s long and deep use of the poets.
It is the setup, the theoretical framings of the first half that need the most work. The author goes through material that to modernist scholars are now old hat (like line 150: why did Eliot admire the Metaphysicals? etc.). But most crucially, the area around line 169 feels like it’s writing Paul Fussell’s classic The Great War and Modern Memory (1975). Fussell is a glaring absence here, as he is the one who has already answered many of the questions that the author here is posing as if for the first time. Fussell’s work being now fifty years old, the author should also brush up on the major critiques and responses to his work.
Much of the first half seems to take a while, so reducing it would allow the author to more quickly get to the strong points of the second half. For example, the point of lines 263-316 seem unclear until we get to line 318, and this whole section could be presented quicker. Also, since this article is for an issue of Humanities dedicated solely to Ford, the background to Ford studies stuff like lines 197-214 should be condensed, and things like footnote 7 should go.
Overall, this is a promising piece of scholarship that has the advantage of powerfully combining the critical debates of two separate eras. What it needs is better framing about how the poetry of the war used its inherited traditions.
Some local issues:
41 – the referenced works at the end list both the Vintage Parade’s End and two of the Carcanet editions—it might make sense to choose only one edition instead of both, preferably the Carcanet
52 – the quote is “astonishingly” not “astonishing”
88 – I’m not sure where footnote 8 is supposed to be in reference to
227 – this quote in mangled
367 – separate author and speaker in “Prufrock”
Typos in lines 81, 86, 91, 134, 167, 168, 204, 296, 320, 329, 364, 466, 492
Author Response
Thank you so much for all the detailed feedback; attached is my description of how I've incorporated your suggestions.

Round 2
Reviewer 2 Report
Comments and Suggestions for Authors
I now recommend full publication based on the revisions. The theoretical apparatus is now more efficient and useful, allowing your major points and good close readings to be persuasive.
Author Response
Great, thank you for all your feedback and pointing me to Fussell!